# Retinogenesis of the Human Fetal Retina: An Apical Polarity Perspective

**DOI:** 10.3390/genes10120987

**Published:** 2019-11-29

**Authors:** Peter M.J. Quinn, Jan Wijnholds

**Affiliations:** 1Department of Ophthalmology, Leiden University Medical Center, 2300 RC Leiden, The Netherlands; pq2138@cumc.columbia.edu; 2The Netherlands Institute for Neuroscience, Royal Netherlands Academy of Arts and Sciences, 1105 BA Amsterdam, The Netherlands

**Keywords:** apical polarity, crumbs complex, fetal retina, PAR complex, retinal organoids, retinogenesis, gene augmentation, adeno-associated virus (AAV), Leber congenital amaurosis

## Abstract

The Crumbs complex has prominent roles in the control of apical cell polarity, in the coupling of cell density sensing to downstream cell signaling pathways, and in regulating junctional structures and cell adhesion. The Crumbs complex acts as a conductor orchestrating multiple downstream signaling pathways in epithelial and neuronal tissue development. These pathways lead to the regulation of cell size, cell fate, cell self-renewal, proliferation, differentiation, migration, mitosis, and apoptosis. In retinogenesis, these are all pivotal processes with important roles for the Crumbs complex to maintain proper spatiotemporal cell processes. Loss of Crumbs function in the retina results in loss of the stratified appearance resulting in retinal degeneration and loss of visual function. In this review, we begin by discussing the physiology of vision. We continue by outlining the processes of retinogenesis and how well this is recapitulated between the human fetal retina and human embryonic stem cell (ESC) or induced pluripotent stem cell (iPSC)-derived retinal organoids. Additionally, we discuss the functionality of in utero and preterm human fetal retina and the current level of functionality as detected in human stem cell-derived organoids. We discuss the roles of apical-basal cell polarity in retinogenesis with a focus on Leber congenital amaurosis which leads to blindness shortly after birth. Finally, we discuss Crumbs homolog (*CRB*)-based gene augmentation.

## 1. The Physiology of Vision

Vision is perhaps the most dominant sense in daily life and both non-correctable unilateral and bilateral vision loss severely impact the quality of life [1]. Vision begins with the processing of light, which is electromagnetic radiation that travels as waves (Figure 1A). Light waves, as with all waves, can be characterized by their wavelength (distance between wave peaks), frequency (number of wavelengths within a time period), and amplitude (the height of each peak or depth of each trough). Visible light is a narrow group of wavelengths between approximately 400 nm and 760 nm which we interpret as a spectrum of different colors (Figure 1B) [2]. Light can be reflected (bounce of a surface), absorbed (transfer of energy to a surface), or refracted (bending of light between two mediums) (Figure 1C).

When light first enters the eye, it is refracted by the cornea through the pupil, whose size is controlled by the iris. The iris, the colored part of the eye, controls the amount of light entering the eye while the lens focuses the light through the vitreous humor and on to the proximal surface of the retina (Figure 2A). The adult retina consists of one glial cell type, the Müller glial cells, and six major types of neurons, the rod and cone photoreceptors, bipolar cells, amacrine cells, horizontal cells, and ganglion cells (Figure 2B). Their cell bodies are distributed across three nuclear layers, the outer nuclear layer (ONL), inner nuclear layer (INL), and ganglion cell layer (GCL). Two synaptic layers, the outer plexiform layer (OPL) and inner plexiform layer (IPL), contain the axonal and dendritic processes of the cells [3]. Whereas there is one type of rod photoreceptor, there are various subtypes of cone photoreceptor, bipolar, amacrine, horizontal, and ganglion cells that differ in their functional roles and morphology [4]. Besides Müller glial cells there are two other glial cell types that serve to maintain retinal homeostasis, the astrocytes and resident microglia [5]. Light must be channelled through the retina and absorbed by its three light responsive cells: the rod and cone photoreceptors and the intrinsically-photosensitive retinal ganglion cells (ipRGCs). The mammalian retina contains various opsin proteins involved in the photoreception synchronisation of circadian rhythms (photoentrainment). These are the cone opsins (M/LWS, red/green opsin; SWS1, blue opsin) responsible for high visual acuity, resolution, and color vision (photopic vision), and rod opsin (RH1, Rhodopsin) responsible for dim light vision (scotopic vision) and ipRGCs opsin (OPN4, Melanopsin) responsible for synchronisation of the circadian rhythms and ambient light perception [6,7,8,9]. The cones are less sensitive to light and rods are more sensitive to light and are also used together under intermediated light conditions (mesopic vision) [10]. Most forms of inherited retinal disease negatively affect the function of photoreceptors, resulting in progressive loss of rod and/or cone photoreceptors. Müller glial cells mediate the channelling of light through the retina towards the photoreceptors [11,12]. Müller glial cells can channel different wavelengths of light to specific subsets of photoreceptors to optimise day vision [13]. The visual pigments of the photoreceptors contain an opsin protein covalently linked to the chromophore 11-*cis*-retinal. Upon the absorption of a photon 11-*cis*-retinal becomes isomerised to all-*trans*-retinal, this leads to an activated opsin intermediate (metarhodopsin II, rods; Meta-II, cones). This active intermediate leads to triggering of a transduction cascade resulting in hyperpolarisation of the photoreceptors, due to the closure of cyclic guanosine 3′,5′-monophosphate (cGMP)-gated channels, and a reduction in glutamate release [14]. This electrophysiological signal is then further propagated to the inner retina and can be propagated through many different pathways to the ganglion cells. Prototypically these signals can be direct, from photoreceptor (PRC) to bipolar cells to ganglion cells. However, it can also be indirect with lateral modulation of the electrophysiological signals being made by horizontal cell processes in the OPL or by amacrine cell processes in the IPL [10,15,16]. Thus, creating radially aligned “functional units” of photoreceptors, bipolar cells, amacrine cells, horizontal cells, and ganglion cells. The fovea contains a specialised pathway, termed the midget pathway, which helps account for its ability to provide high visual acuity [17,18,19].

The visual system, however, is not solely comprised of the eye but also the topographically mapped ganglion cell axonal projections connecting the retina to the superior colliculus (SC) and lateral geniculate nucleus (LGN) in the brain [20]. The ganglion cell axonal projections exit the left and right eyes as bundles, the optic nerves, and they extend to below the hypothalamus to the optic chiasm. The optic chiasm is the crossover point for the nasal axons of each eye which combine with the opposing eyes temporal axons. The two optic tracts extend from the optic chiasm to the SC and the LGN, with the optic radiations further extending from the LGN to the primary visual cortex (Figure 2C) [21]. The SC, LGN, and pulvinar nuclei are all involved in the process of relaying and refining visual information to the primary visual cortex [22,23]. Interestingly, despite the severe retinal dysfunction of Leber congenital amaurosis-2 (LCA2) patients, recovery of both retinal function, but also reorganization and maturation of synaptic connectivity in the visual pathway, is found upon administration of a gene therapy treatment [24]. Such recovery highlights the relative plasticity of the human visual system.

## 2. Retinogenesis

The retina, part of the central nervous system, offers an extremely accessible and relatively immune-privileged model system for investigating the mechanisms of neural development and vision [25]. A high conservation of the genes involved in retinal development exists across species allowing us to gain an in-depth fundamental knowledge of these mechanisms. Retinal development is both a pre- and postnatal process. The development of the retina begins when the anterior neural plate subdivides into a number of domains, with the medial region specifying as the eye field (Figure 3). The formation of the eye field is coordinated by expression of the eye field transcription factors (EFTFs), shortly after gastrulation. There are a number of EFTFs in mammals including Pax6, Rax, Lhx2, Six3, and Six6. The eye field consists of all the progenitors which go on to form all the neural-derived cell types and structures of the eye [26,27,28,29,30]. The progenitors of the eye field begin to specialize very early in development, hence the large number of bilateral diseases of eye morphogenesis [28]. From the eye field, bilateral optic sulci form and evaginate from the diencephalon at human fetal embryonic day 22 (E22) forming optic vesicles at E24 (Figure 3). The optic vesicles extend towards the surface ectoderm remaining connected to the forebrain through the optic stalk, which eventually develops into the optic nerve. The hyaloid artery, running from the optic stalk and into the retinal neuroepithelium through the optic fissure, provides the basis for the vascularisation of the retina and developing eye. As the optic vesicles invaginate forming the two-layered optic cups by E32, the surface ectoderm thickens forming the lens placode and further develops into lens vesicle, sitting behind the surface ectoderm (Figure 3). The anterior rim of the optic cup will become the iris and ciliary body, while the posterior rim will become the pigmented and neural retina. The outer layer of the posterior optic cup remains as a single cuboidal layer becoming the retinal pigment epithelium (RPE). The single inner layer of the posterior optic cup proliferates and differentiates, beginning in the 7th fetal week (Fwk), developing into the multilayered neural retina [28,31].

The processes of the newborn progenitors of the inner optic cup, the retinal neuroepithelium, extend and attach both apically through adherens junctions (AJs) at the outer limiting membrane (OLM), and basally through integrin- and proteoglycan-based focal adhesions at the inner limiting membrane (ILM) [32,33]. Retinal progenitors undergo interkinetic nuclear migration in which their nuclei move in an apical-basal manner in phase with the cell cycle, this occurs in mainly a stochastic manner but becomes briefly directed at cell division (Figure 4A) [33,34,35]. Progenitors initially undergo symmetric cell division leading to an increase in the pool of progenitors and thus thickening of the neuroepithelium. After that the progenitors go through asymmetric divisions, and produce one daughter cell to maintain the stem cell pool and one terminally differentiated postmitotic cell. Later in development depletion of the retinal progenitor pool occurs through symmetric divisions leading to two postmitotic terminally differentiated daughter cells (Figure 4B) [34,36]. Cell intrinsic and extrinsic factors govern cell fate choice and thus tissue architecture and function. The retinal cells governed by these factors progress from multipotent retinal progenitors to competent postmitotic precursors, which undergo further specification before becoming the final differentiated adult cell type [4,37,38,39].

The birth of the seven major cell types of retina occur from the early multipotent retinal progenitor cells and happens in an orderly and overlapping manner [39]. The genesis of the major cell types group into an early phase and a late phase. The early phase consists of the birth of the first ganglion cells, cone photoreceptors, horizontal cells, and amacrine cells. The overlapping late phase consists of the birth of the first rod photoreceptors, Müller glial cells and bipolar cells (Figure 4C) [39]. Recently, both Aldiri et al. (2017) and Hoshino et al. (2017) described similar retinal time courses for the developing human retina based on RNA-Seq analysis [40,41]. The newborn postmitotic cell types must become positioned correctly within the retina; this occurs through migration of cells along the radial axis (apical-basal) of the retina or by tangential migration of cells perpendicular to the radial axis of the retina. Interestingly, only the early born cell types (ganglion cells, cone photoreceptors, horizontal cells, and amacrine cells) exhibit tangential migration (Figure 5A) [42,43].

There are a number of modes of radial migration for newborn neurons including: glial cell-guided, the migration of neurons along radial glial progenitors (Figure 5B); Somal translocation, the movement of nuclei across inherited apical or basal processes (Figure 5C); Multipolar migratory mode, nuclei movement due to multiple cell processes with no retention of apical or basal attachment to facilitate nuclei movement (Figure 5D); No translocation, inefficient migration due to retention of the apical or basal process and slow release of opposing process (Figure 5E). These various modes of migration are cell type-specific [44,45,46,47]. Tangential dispersion is driven by a mix of diffusible signals and/or contact-mediated interactions that drive a local spacing rule to keep a minimum distance between neighboring cells of the same cell type [48].

Retinal mosaic is the term used for the distribution of a neuronal cell type orthogonal to the apical-basal axis in a particular retinal layer. There is a highly ordered mosaic architecture in the mammalian retina leading to the non-random distribution of its cell bodies and dendritic process. This mosaic patterning is essential for retinal functionality, tying information together in a regularly patterned/ordered way from radially aligned “functional units” such that complete sampling and coverage of an image is achieved. Development of mosaics may be due to a combination of tangential dispersion (for early born cell types), programmed cell death, and lateral inhibition [45,48,49]. Interestingly, mosaic patterning can apply to a group of cells that have yet to reach their final developmental position, suggesting a pre-orchestrated cell intrinsic process [45].

Thus, retinogenesis is a precise orchestration of spatiotemporal processes such as symmetric and asymmetric cell division, cell fate choice (determination, competence, specification, and differentiation), cell migration (interkinetic nuclear migration, radial migration, and tangential migration), and maturation (integration and specialization of retinal spatiotemporal processes to provide adult functionality). The developing retinal neuroepithelium has a large amount of plasticity to accommodate these spatiotemporal process while maintaining its tissue integrity and architecture.

## 3. The Genetics of Retinal Development

As briefly highlighted in the previous section a number of genes are responsible for forming the early eye field. In this section we will shortly expand on some of the important gene regulatory networks (GRNs) essential for retinal development. GRNs can establish precise spatial, temporal, and cellular context specific controlled changes in gene expression patterns through the synergistic relationship of sets of transcription factors (TF) and their action on cis-regulatory modules (CRMs). The CRMS typically are a collection of TF-binding sites on the same strand of DNA as they affect [50,51,52,53]. GRNs are important as they can provide us with mechanistic insight into what is need to acquire and maintain a particular cell type identity. We will discuss the GRNs responsible for retinal progenitors and subsequent competent postmitotic precursors and their cell type specification. Mutations in, or misregulation of, several of these early developmental genes can lead to inherited retinal diseases. A number of recent papers have focused on retinal development using bulk transcriptomic [40,41,54,55] and single-cell transcriptomic approaches [54,56,57,58,59,60,61] to study human fetal retina and retinal organoids. These works add too many of the findings from work on mammalian animal models which have defined developmental or cell specific gene clusters and networks [62,63,64,65,66,67,68,69,70].

Several genes have been attributed to neuroretinal specification as well as the proliferative and multipotent ability of retinal progenitor cells, including *Vsx2* (also known as *Chx10*), *Pax6*, *Lhx2*, *Rax*, *Six3,* and *Six6* [71,72,73,74,75,76,77,78,79]. Many of the genes are also implicated in retinal abnormalities; for instance, *Pax6* mutations can lead to foveal hypoplasia, while *Rax* mutations can cause microphthalmia leading to retinal dysplasia [80,81]. Two genes, *Ikzf1* and *Casz1,* are required for the temporal regulation of retinal progenitor cell fate, with dysregulation of these genes leading to changes in the production of early versus late-born retinal cell types [82,83]. Interestingly, many retinal progenitor cell transcription factors are also important in Müller glia cell specification [68]. This includes the Hippo effector Yap, which is essential for retinal progenitor cell cycle progression. Additionally, Yap is required for Müller glial cell reprogramming and cell cycle re-entry and is misregulated in retinal disease [84,85,86,87]. Other factors related to retinal progenitors and Müller glial cells include Notch factors Hes1 and Hes5 as well as Lhx2, Rax, and Sox9 [88,89,90,91].

Several retinal TFs including Otx2, Crx, Nrl, and Nr2e3 control rod and cone-specific photoreceptor specification. Mutations in *Crx* can cause Leber congenital amaurosis (LCA), cone-rod dystrophy (CRD), and Retinitis pigmentosa (RP), while *Nrl* and *Nr2e3* mutations can cause RP and enhanced S-cone syndrome [92,93,94,95,96,97,98]. Otx2 can determine both rod and cone photoreceptor cell fate, while Crx acts with Nrl and Rorβ for terminal photoreceptor gene expression controlling the cone/rod ratio [99,100,101,102]. Activation of *Nrl* expression leads to the subsequent activation of the *Nr2e3* rod-specific factor; both Nrl and Nr2e3 can suppress cone cell fate genes [101,103,104]. Prdm1 (also known as Blimp1) also promotes rod specification while repressing bipolar fate [105,106]. Thrβ2 and RXRgamma are required for cone generation and subtype specification [107,108,109]. A CRM of the *Thrb* gene is regulated by Otx2 and Onecut1 transcription factors for the production of cones and horizontal cells, with Onecut1 found to be critical in specifying cone versus rod fate [110]. Recently, the Emerson Lab further confirmed that ThrbCRM1 progenitor cells preferentially form cone photoreceptors as well as subtypes of horizontal and ganglion cells [111].

Bipolar cells are also specified from Otx2 component postmitotic precursors in which expression with Vsx2 leads to their cell specification [105,106]. Vsx1 and Bhlhb5 are required for bipolar cell subtype fate [112,113]. The other interneurons, amacrine cells, and horizontal cells arise from Pax6, Ptf1a and Foxn4 expressing retinal progenitor cells [76,114,115]. Prox1 lies further downstream of Foxn4 and Ptf1a and specifies horizontal cell fate [116]. While, Onecut1 acts downstream of Foxn4, in parallel with Ptf1a, but upstream of Prox1 to specify horizontal cell fate [117]. Additionally, Lim1, Isl1 and Lhx1 also specify horizontal cell fate [118,119,120]. Tfap2a and 2b, Barhl2, Bhlhb5, NeuroD factors, and Isl1 act downstream of Ptf1a to specify an amacrine cell fate [113,121,122,123,124]. Lastly, Pou4f2 and Isl1 are essential in the acquisition of ganglion cell fate being downstream of retinal progenitor cell factor Atoh7 [125,126]. Additionally, genes promoting ganglion cell specification include *Neurod1*, *Sox4,* and *Sox11* [127,128].

Nevertheless, what has been found out about *CRB1* transcript expression in early retinal development? Recently, Hu et al. found using single-cell RNA-seq that *CRB1* transcripts were particularly enriched during human retinal development in retinal progenitor and Müller glial cells from human fetal retina [56]. In human retinal organoids, *CRB1* transcripts were found to be lowly expressed in very early organoids with moderate expression in later organoids [57]. In a study by Clark et al. they found using single-cell RNA-seq that transcripts for *Crb1* in mouse retina increased from embryonic to postnatal stages. Interestingly, they found the opposite for *Crb2* transcripts, being more abundant early embryonically and decreasing postnatally [63]. This pattern is in agreement with studies of human fetal retina and retinal organoids that show initial low protein levels of CRB1 and higher levels of CRB2 in early development [129]. Redundancy of function for CRB1 and CRB2 has been identified in the mouse retina. With knockout of either *Crb1* or *Crb2* in mouse Müller glial cells leading to mild retinal morphological phenotypes, while ablation of both *Crb1* and *Crb2* concomitantly from mouse Müller glial cells leads to a severe Leber congenital amaurosis phenotype [130,131,132]. However, while *Crb1* knockout mice are viable, *Crb2* ablation causes lethality in mice and genetic variants of the *CRB2* gene cause a syndromic phenotype in humans [133,134]. CRB3 is more widely expressed in epithelial tissues than either CRB1 or CRB2 and does not share their large extracellular domains. Ablation of *Crb3* leads to lethality shortly after birth with defects in (lung and intestinal) epithelial tissue morphogenesis [135,136].

## 4. Morphological and Molecular Recapitulation of the Human Fetal Retina

Studying retinal development in animal models is useful to understand the underlying cellular processes, but some of these processes are remarkably different in the human retina. There are molecular and morphological similarities between mammalian retina, but subtle differences highlight caution when applying information from other species to the human retina and its diseases, even between non-human primate and human retina [137,138,139,140]. Additionally, limited access to human fetal and adult cadaveric donor retina limited the studies in human retina at a tissue or single cell type level in regards to pathobiology, signaling dynamics, and physiology. Retinal organoids derived from human embryonic stem cells or induced pluripotent stem cells mimic the three-dimensional laminated structure of the retina allowing us to study basic eye development, disease modelling, drug potency assays, gene augmentation, and cell therapeutic strategies [141,142,143,144,145,146].

Multiple recent studies have directly compared human fetal retina with human stem cell-derived retinal organoids, analyzing their morphology or transcriptome, or both [54,147,148,149]. Additionally, a large body of research exists regarding the morphology and transcriptome of either human fetal retina [8,40,41,150,151,152,153,154,155,156,157,158,159,160] or human stem cell-derived retinal organoids [58,141,142,161,162,163,164,165,166,167,168,169,170]. Together, these studies provide a unique insight into human retinogenesis and allow us to have a reference point between in vivo human retinal development and in vitro models of it. Do human stem cell-derived retinal organoids sufficiently recapture the cell type diversity including all sub-types, morphological cues, synaptic wiring, and light sensitivity of the human fetal retina and subsequent adult retina?

Human retinal organoids have been found to recapitulate the main temporal and spatial cues of the in vivo human retina producing all the retinal layers and containing the 7 major cell types [142,145,171]. The spatiotemporal birth and maturation of retinal cell types differ depending on their location within the retina. In the human fetal retina, photoreceptors are born and remain located in the apical retina. However, in human stem cell-derived retinal organoids, photoreceptors reside mostly apically but are also seen in more basal positions particularly in early developmental stages, suggesting incomplete positional cues. Interestingly, Kaewkhaw et al. (2015) were able to track basally located photoreceptors from DD42 to DD44 with live cell imaging, finding that they translocated their soma apically [168]. The mammalian retina matures in a centroperipheral manner with the peripheral retina organizing its adult like lamination later than the central retina [155]. This delayed peripheral maturation is also detected in gene expression profiles comparing these regions. Cell type expression patterns, analyzed by immunohistochemistry, can be delayed by at least 50 days in the peripheral compared to more central retina. Differences are also found between nasal and temporal retina [41]. Additionally, sustained differences in gene expression profiles of peripheral and macula of the human adult retina is found, but this has been attributed to the anatomical differences between these two regions [172]. Mammals are known to contain a peripherally located ciliary marginal zone (CMZ), a stem cell-like niche able to contribute postmitotic cells to the retina in addition to radial glial progenitor cells [173,174]. However, little data is available on the CMZ of the human fetal retina, but studies on human and mouse stem cell-derived retinal organoids have highlighted its presence [164,165].

The human retina contains a central region termed the macula in which the 1.5 mm fovea centralis resides. The fovea, required for high visual acuity and color vision, in particular, was found to develop and maturate much earlier than the rest of the retina [41,175]. The developing fovea, already free of mitotic cells at Fwk 8.4, initially consists of a single row of cone photoreceptors. The rapidly maturating foveal inner retina already consists of the main components of the midget ganglion pathway as well as of presynaptic markers present in the OPL and IPL by Fwk 13.7 [41]. However, a small depression called the foveal pit develops from Fwk 25 which leads to a gradual receding of all retinal layers except for the outer nuclear layer which begins to thicken. Foveal development continues in early childhood with elongation of cone outer segments and is considered adult-like in early- to mid-adolescence [152,153,160,176,177]. Newborns still have underdeveloped outer segments at the cone-rich fovea after birth. Therefore, suggesting that human newborns rely on extrafoveal vision initially [138]. Immature cones are present from Fwk 8 and immature rods from Fwk 10. Cone S opsin expresses from Fwk 12 while cone L and M opsin and rhodopsin express at Fwk 15 [8,151,153,155,159]. Rod photoreceptor maturation defined as the development of both inner and outer segments, occurs quicker in the mid-peripheral human retina than the parafoveal region [150]. Full maturation of the human retina does not occur until early adolescence. The average adult human retina contains 4.6 million cones, decreasing sharply in density outward from the fovea. However, the average number of rods in the adult human retina, 92 million, far exceeds the number of cones. The density of rods is highest around the optic nerve and decrease in density towards the peripheral retina and are absent from the central fovea [156].

Retinal organoids do not yet form a separate specialized fovea. However, they do form all of the main photoreceptor subtypes including photoreceptors with Rhodopsin, L/M-opsin and S-opsin [146,178,179]. Retinal organoid photoreceptors form both inner and outer segments and contain many of the prototypic morphological structures including: mitochondria, basal body, centriole, connecting cilium, and outer segment discs. Additionally, adjacent structures, such as the adherens junctions of the OLM, the microvilli of radial glial progenitors, and subsequently Müller glial cells are also found [142,162,163,169,170,171,179]. The quality of these structures greatly varied between studies and in particularly outer segments had very few or disorganized disk stacks. Recently, Eldred et al. (2018) found that the DD200 retinal organoids can recreate the temporal generation of cones having a similar distribution, gene expression profile, and morphology as adult human retina. They also identified a temporal switch which promotes the differentiation of S cones towards L/M cones through thyroid hormone signaling. Going one step further they also found that the two thyroid hormone states, T3 and T4, were modulated by different deiodinase enzymes in early or late organoids to specify the change from S to L/M cone subtypes and found that temporal gene expression of these enzymes correlated with data from Hoshino et al. (2017) on the human developing retina [41,146]. Interestingly, preterm human infants that have low T3/T4 are more likely to have defects in color vision [180]. Local modulation of thyroid signaling in retinal organoids may specify a more cone-rich, fovea-like region, making them a step closer to the in vivo human retina. Such modulation of thyroid signaling in retinal organoids might be achieved through optogenetically controlled protein expression. Recently, Kim et al. were able to make more cone-rich retinal organoids by using an improved differentiation protocol [57].

## 5. Light Responsiveness and Synaptic Transmission of the Fetal Retina and Stem Cell-Derived Retinal Organoids

Light responsiveness and synaptic transmission are critical milestones for a functioning retina. As previously mentioned, the fetal retina does have both rods and cones with developing inner and outer segments as early as Fwk 15. The fetal retina is however considered not fully mature until post birth. The eyelids open by approximately Fwk 25, a milestone that suggests that the fine tuning of the retina to visual stimulation can begin from this point [181]. In utero visual stimulation is considered very limited. However, fetal responses to visual stimulus as measured by heart rate, physical movement and brain activity have been reported. Transabdominal illumination has been reported to increase fetal heart rate at Fwk 36 (Smyth et al. 1965: Fwk not reported, Kiuchi et al. 2000: Fwk 36–40) and fetal movement at Fwk 26 [182,183]. Fetal heart rate increase has also been shown during amnioscopy in which a cold light source was exposed to the amnion and fetus for 30 s at Fwk 38 [184]. An increase in fetal heart rate in response to light with increasing gestational age as analyzed by actocardiogram is seen from Fwk 18 to 41 [185]. However, fetuses have only been reported to reliably respond to light stimulation from Fwk 37; this is likely due to differences in abdominal and uterine wall thicknesses, light sources used, the distance of light source, and its focus on the fetus eyes [185,186,187]. Magnetoencephalography (MEG) studies have recorded visually evoked brain activity from as early as Fwk 28 [188,189,190,191]. Functional magnetic resonance imaging (fMRI) has also been successfully used to measure fetal response to transabdominal illumination, finding activity in the frontal lobes but not the visual cortex from Fwk 36 [192].

Two main techniques have been utilized to measure preterm neonatal vision, the electroretinogram (ERG) for retinal activity and visual evoked potentials (VEP) for brain activity. VEP studies suggest that extrauterine age accelerates the development of the fetal visual system once a maturational threshold has been reached at post Fwk 25 [193,194,195,196]. Interestingly, this coincides with the approximate age of eyelid opening. Similarly, from as early as Fwk 31, ERG studies on preterm neonates suggest that improvements in retinal activity are correlated with postconception age and extrauterine age [196,197,198]. Improvements in retinal activity can be recorded by a decrease in a- and b-wave latency with an increase in amplitude. Preterm ERG studies cannot be used to assess if in vivo retinal activity also has a maturational threshold because the eyelids are yet closed before Fwk 25. Other features of preterm ERG include decreasing rod threshold with increasing postconception age, adult-like b-wave sensitivity is reached at six months after normal birth [199,200]. Rod functional maturation occurs peripherally and then parafoveally [201]. Preterm birth, however, has been consistently linked with reduced rod and cone function when compared to usual term infants [202,203,204]. This suggests that premature exposure of the retina to light is harmful. It is unknown if premature light exposure harms cultured stem cell-derived retinal organoids.

As previously mentioned, human retinal organoids develop photoreceptor outer segments of different quality between week (Wk) 16-28 [142,146,162,163,167,169,170,171,179,205]. Retinal as well as brain organoids have been shown to respond to light [142,162,206]. Zhong et al. (2014) found that 2 out of 13 rod cells with putative outer segments in Wk 25-27 hiPSC-derived retinal organoids responded to light as measured by perforated-patch recordings in the voltage-clamp mode. The sensitivity of these immature human rod cells was less than found in adult non-human primate photoreceptors. Multiple responses could not be elicited in these retinal organoids, likely due to a depletion in components required for phototransduction [142]. Hallam et al. (2018) used a 4096 channel multielectrode array (MEA) on which they flattened longitudinally opened Wk 21.4 (DD150) hiPSC-derived retinal organoids ganglion side down and detected changing spike activity from pulses of white light. They also used puffs of cGMP, which depolarizes photoreceptors leading to an unstimulated condition (Dark Current), to show that the light responses found were driven by phototransduction in photoreceptors and not the potential activity of ipRGCs [162]. Quadrato et al. (2017) found that eight month hiPSC-derived brain organoids exhibited spontaneously-active neuronal networks, using high-density silicon microelectrodes. Additionally, subpopulations of neurons were identified which were responsive to 530 nm light in 4 out of 10 organoids. However, they were unable to attribute the responses directly to the photosensitive cells or the downstream neuronal networks [206].

Human stem cell-derived retinal organoids can develop some synaptic maturity. Electron microscopy data showed the presence of electron-dense ribbons surrounded by synaptic vesicles. Immunohistochemistry markers such as PSD-95, vGlut1, PNA, Synaptophysin, and Syntaxin 3, confirm synapses in human stem cell-derived retinal organoids. Markers such as RIBEYE (as detected by CtBP2) and Bassoon confirm the presence of ribbon synapses. These markers roughly aligned at the outer plexiform layer of retinal organoids. However, this varied between protocols and degeneration of the inner retina in ageing organoids may also play a factor [148,162,167,170,171,178]. Interestingly, Dorgau et al. (2018) showed that blocking of the extracellular matrix protein Laminin γ3 in late-stage retinal organoids led to the disruption of ribbon synapse marker Bassoon. This may be due to the significant disruption of Müller glial cell end feet at the inner limiting membrane [148]. Wahlin et al. (2017) showed that both excitatory (L-aspartate, glutamate) and inhibitory (GABA, glycine) neurotransmitters of the retina were present in Wk 43 (DD300) hiPSC-derived retinal organoids. L-aspartate was found in the ONL, glutamate and GABA throughout the retina and glycine in the INL [170]. Hallam et al. (2018) used puffs of GABA to highlight the emerging functional neural networks in Wk 21 (DD150) human stem cell-derived retinal organoids [162]. Wahlin et al. (2017) used whole-cell patch clamp recordings to elicit membrane capacitance changes as an index of voltage-dependent synaptic vesicle release in retinal organoid photoreceptors [170]. Similarly, Deng et al. (2018) were able to show the electrophysiological response from whole-cell patch clamp recorded rod photoreceptors from hiPSC-derived retinal organoids [178].

Together, these data highlight the potential for producing light responsive human stem cell-derived retinal organoids with maturating synapses that are capable of transferring information from the photoreceptors to the inner retina. At the least, retinal organoids have “functional units” of photoreceptors, bipolar cells, and ganglion cells. Work needs to be carried out to assess the quality of these neural networks and how well all cell types contribute. More emphasis needs to be put on comparing the early in vivo and in vitro retinal response to light and how that ties in with the maturational stage. Unsurprisingly, retinal organoids represent an immature/incomplete development stage, and thus functional responses reflect this. Improved culturing methods and better control of the microenvironment might help in further maturation of the retinal organoids.

## 6. Improved Retinal Organoid Modelling

Despite ongoing issues with batch to batch variations, many of the current differentiation protocols lead to the generation of well laminated retinal organoids that contain all the primary retinal cell types but have putative photoreceptor segments or show in long-term cultures degeneration of the inner retina. This may be due to the lack of correct microenvironmental cues and structural support. Degeneration of the inner retina is likely due to the lack of access by medium components, particularly as the retina thickens during development. Furthermore, misregulation of ECM components in human stem cell-derived retinal organoids may significantly affect their correct lamination [148]. Reproducibility and staging of human retinal organoids is also an important consideration particular as we further explore their use in developmental studies and disease modelling [207]. Many teams are focusing on new methodologies to improve the quality of retinal organoids by tweaking protocols, using bioreactors, or microfluidic systems, [163,167,169,170]. Recently, Mellough et al. (also with slight modification performed by Cowan et al.) showed that mechanical dissociation at the embryoid body formation stage lead to improved formation of human retinal organoids [208,209]. The use of bioreactor setups, instead of static culture setups, may help solve a number of these problems as they allow for improved aeration and distribution of nutrients as well as allow for scaling up of organoid production. Bioreactor setups report stem cell-derived retinal organoids with better lamination and enhanced differentiation, an increased yield of photoreceptors with outer segment structures, improved cone formation and a better recapitulation of the spatiotemporal development of in vivo retina [167]. While the initial stages of differentiation require hypoxic conditions, improved oxygen diffusion at later stages is essential for greater cell proliferation and ganglion cell survival. In the absence of vascularized stem cell-derived retinal organoids, bioreactors help facilitate this mechanism. A number of teams have also very recently come up with methods that lead to better cone and rod specification in human retinal organoids [57,210,211,212,213].

Many of the differentiation procedures used to derive human retinal organoids lead to the concomitant production of RPE. However, it produces RPE that is consistently not directly adjacent to the photoreceptor segments when using a 2D to 3D differentiation protocol. It does allow however medium that is conditioned by both the retinal organoids and RPE, which may provide some essential diffusible factors for both structures. Production of full-length photoreceptor segments by Wahlin et al. (2017) suggested that contact of RPE is non-essential for their development/maturation [170]. However, correctly located RPE may provide essential structural support and may help to facilitate a number of the physiological roles of photoreceptor segments such as phototransduction. Microfluidic systems for retinal organoids may help to promote improved cell to cell interaction and additionally provide tighter control of the microenvironment [214,215]. One of the future uses of human stem cell-derived retinal organoid technology is potentially as a source of transplantable tissue and in particularly photoreceptor cells to treat retinal diseases [171,216,217,218]. Integration and functionality of transplanted photoreceptors into host retina has been shown to be much more limited than initially thought, with predominant cytoplasmic material transfer including fluorescent reporter proteins [219,220,221]. Transplanted photoreceptors may not facilitate their physiological roles fully due to lack of interaction with the host RPE. One way to enhance the photosensitivity of transplanted photoreceptors from human stem cell-derived retinal organoids is to use optogenetically transformed photoreceptors [222]. Material transfer is found as well from conjugates formed from transplanted NTPDase2-positive CellTracker Green labelled Müller Glial Cells [223,224]. Material transfer represents a novel route to develop cell-based therapeutics which may be able to transfer “healthy components” to diseased retinal cells [225].

## 7. The Apical CRB and PAR Complexes

Apical-basal cell polarity is pivotal for the formation and functionality of epithelial tissues being governed by conserved canonical factors that define the apical domains. Apical polarity factors are for example the Crumbs-homologues (CRB1 and CRB2), Protein associated with Lin Seven 1 (PALS1 also called MAGUK p55 subfamily member 5 or MPP5), Partitioning defective-6 homolog (PAR6), atypical protein kinase C (aPKC), and PAR3. Basolateral polarity factors are for example Protein scribble homolog (SCRIB), Discs large homolog (DLG), and Lethal giant larvae protein homolog (LGL). However, recent work in the fruit fly *Drosophila* mid gut indicates that there are alternative apical polarizing factors other than the canonical epithelia polarity factors and that this may also extend to some types of vertebrate epithelia [226]. Additionally, there also exists planar cell polarity (PCP) in tissue epithelia, which is orthogonal to the apical-basal axis (Figure 6A) [227]. The retinal sub-apically localized CRB and PAR complexes (Figure 6B) are pivotal in maintaining the spatiotemporal processes of retinogenesis. The CRB complex has a prominent role in the control of apical-basal polarity acting as a sensor for cell density and upon polarization leading to regulation of Adherens Junctions (AJs) to promote maintenance of cell adhesion [228,229]. Disruption of the CRB complex leads to loss of polarity and can lead to subsequent loss of adhesion, ectopic localization of progenitors and postmitotic cells due to disrupted apically anchored process and coordinated cell migration, increase in cycling progenitor cells and late-born cell types, increase in early retinal apoptosis, and disruption of lamination. A long-term consequence of loss of retinal apical polarity is mild to severe retinal degeneration with a concurrent loss of retinal function in line with morphological deficit [130,131,230,231,232,233]. The complex acts in the role of conductor coordinating multiple downstream signaling pathways which have essential roles in development, such as the Notch and Hippo pathways [228,234,235]. Thus, leading to the regulation of cell size, cell fate determination, cell self-renewal, proliferation, differentiation, mitosis, and apoptosis. However, how these intertwined cellular responses are mediated collectively by the core complex remains ambiguous.

The mammalian retinal CRB complex comprises of at least one of the three CRB family members, CRB1, CRB2, and CRB3 (isoform CRB3A which has a conserved C-terminus, isoform CRB3B which lacks the conserved C-terminus), in addition to PALS1 (also called MPP5), PATJ, MUPP1, MPP3, and MPP4. Both CRB1 and CRB2 have a large extracellular domain with epidermal growth factor (EGF) and laminin-globular domains. As discovered in *Drosophila* epithelia cells CRB1, CRB2, and CRB3A have a single transmembrane domain juxtaposing a short intracellular C-terminus of 37 amino acids which contains a FERM-binding motif (4.1, ezrin, radixin, moesin) and PSD-95/Discs-large/ZO-1 (PDZ)-binding motif ERLI (Glu-Arg-Leu-Ile) [144,229,235,236]. The three prototypic CRB proteins are shown in Figure 6C, for further details also on other isoforms details can be found in Quinn et al. 2017 [144]. Alternatively, the non-prototypic CRB3B isoform, which has a role in ciliogenesis and cell division, contains a C-terminal PDZ-binding motif CLPI (Cys- Leu-Pro-Ile) that does not interact with the PAR complex as found in Madin-Darby canine kidney epithelial cells (MDCK) [237,238]. The FERM motif can bind to proteins such as EPB4.1L5 which plays a role in epithelial-to-mesenchymal transition during gastrulation in mice and similar proteins are negative regulators of photoreceptor size in *Drosophila* and zebrafish [239,240,241,242,243]. EPB4.1L5 oligomerization is essential for its binding to CRB and is mediated through its FERM and FERM adjacent (FA) domains, as found in *Drosophila* and in MDCK cells [244]. EPB4.1L5 controls the actomyosin cytoskeleton at both apical junctions and basal focal adhesions, as found in the mouse kidney and during mouse development [239,245]. EPB4.1L5 is predominantly located basolaterally in early development, repressing CRB, but is recruited by CRB apically at later stages of differentiation. In the adult mammalian retina EPB4.1L5 has been found to localize to the OLM [242,243]. PAR complex member aPKC can bind and phosphorylate the FA domain of EPB4.1L5 leading to the dismantling of the EPB4.1L5 oligomer. This phosphorylation by aPKC prevents the premature apical localization of EPB4.1L5, in turn EPB4.1L5 restrains aPKC signaling thus antagonizing each other, leading to tightly controlled segregation of apical/basal membrane domains, as found in *Drosophila* and in MDCK cells [244,246]. The 4 amino acid PDZ-binding motif ERLI of CRB allows for interaction with adaptor proteins such as PALS1 and PAR6, as found in *Drosophila* and in MDCK cells [247,248]. Binding of PALS1 to the C-terminal PDZ domain of CRB leads to recruitment of PATJ or MUPP1 through binding of the PALS1 N-terminal L27 domain to the L27 domain of PATJ or MUPP1, as found in *Drosophila* and in MDCK cells [249,250]. Additionally, PALS1 can recruit MPP3 and MPP4 to the apical complex, in the mouse retina [251,252]. PALS1 is abundantly expressed at the OLM and tight junction of the RPE. Ablation of PALS1 in the mouse neural retina resulted in late onset retinal degeneration suggestion redundancy of MPPs in the neural retina, whereas ablation of PALS1 in the mouse RPE and neural retina results in early onset retinal degeneration suggesting specific roles for PALS1 in RPE [231].

Binding of PAR6 to the C-terminal PDZ binding domain of CRB leads to the recruitment of the other PAR complex members PAR3, aPKC, and cell division control 42 (CDC42). PAR6 can interact with PAR3 through their PDZ domains, with aPKC through their N-terminal Phox and Bem1 (PB1) domains, and with CDC42 through their semi-CDC42- and Rac-interactive binding (CRIB) domains, as found in *Caenorhabditis elegans* and mammalian tissue lysates and cos1 cells [253,254,255,256]. The activity of aPKC is suppressed by PAR6 binding, but this suppression is partially relieved when GTP bound CDC42 interacts with the complex, as found in *Drosophila* and in MDCK cells [257,258]. However, the activity of aPKC has also been shown to be promoted by PAR6 [259]. PAR3 is both an inhibitor of aPKC activity but also its substrate, found in mammalian tissue lysates [256]. At adherens junctions, PAR3 can also bind to the scaffolding proteins FERM domain containing 4A (FRMD4A) and FRMD4B (also known as GRSP-1) leading to the recruitment of cytohesin-1 (CYTH)1 and causing subsequent activation of ARF6, this complex being essential for epithelial polarity [260]. FRMD4B and CYTH3 (also known as GRP1) also exist in a complex, found in COS1 cells [261]. Recently, in the mouse retina a variant of FRMD4B, *Frmd4b^Tvrm222^*, led to the suppression of OLM fragmentation and photoreceptor dysplasia in *Nr2e3^rd7^* and *Nrl^-/-^* mice. Whole exome sequencing revealed that the *Frmd4b^Tvrm222^* variant had a substitution of serine residue 938 by proline (S938P). Transfection of COS7 cells with either S938P or wild-type FRMD4B and the addition of insulin, an agonist of the phosphoinositide 3-kinase (PI3K)-AKT pathway, revealed that the FRMD4B variant does not translocate to the plasma membrane as occurs with wildtype FRMD4B. The *Frmd4b^Tvrm222^* mice showed reduced AKT phosphorylation and an increase in cell junction proteins, activated AKT leads to loss of apical-basal polarity. Therefore, the interactions of the FRMD4B variant and cytohesin-3 may modulate both the PAR3 activated ARF6 pathway and/or PI3K-AKT pathway to prevent retinal dysplasia in *Nr2e3^rd7^* and *Nrl^-/-^* mice [262]. CDC42 in addition to its role in regulating aPKC has recently been shown to regulate another kinase to the apical domain, p21-activated kinase-1 (PAK1), as found in *Drosophila* and mammalian epithelial cells [258,263]. Loss of PAK1 or aPKC leads to a moderate polarity phenotype however a dramatic loss of epithelial polarity was detected when PAK1 as well as aPKC are inactive. Both aPKC and PAK1 act redundantly downstream of CDC42. PAK1 is expressed throughout the mouse retina [263,264]. The CRB and PAR complexes may interact additionally through direct interaction of the amino terminus of PALS1 and the PDZ domain of PAR6, the interaction being regulated by GTP bound CDC42, as investigated in mammalian cell lines [265]. PAR6, PAR3 and aPKC are located at the OLM in the embryonic mouse retina while CDC42 is located throughout the retina [266,267,268].

## 8. The Localization of the Mammalian Retinal CRB Complex

The developing mammalian retina expresses both CRB1 and CRB2. In mouse retina, both CRB1 and CRB2 are expressed at the subapical region adjacent to adherens junctions of progenitor cells at embryonic day 12.5 (Figure 7A), which is equivalent to the early 1st trimester human fetal retinal development, transcriptionally [40,41,232,242,269]. However, in 1st trimester human fetal retina, while CRB2 labelling is found at the subapical region adjacent to adherens junctions in putative photoreceptor inner segments and the apical villi of radial glial progenitor cells, CRB1 is almost below detection level (Figure 7B). Subsequently, in mid 2nd trimester human fetal retina CRB1 and CRB2 labelling could be clearly detected at the subapical region adjacent to adherens junctions between putative photoreceptor inner segments and in the apical villi of radial glial progenitor cells/ Müller glial cells. In addition to CRB1 and CRB2 expression other CRB complex members PALS1, MUPP1, and PAR3 were found to be expressed in 1st and 2nd trimester human fetal retina. In 1st trimester human fetal retina we also found PATJ expression this was not analyzed in 2nd trimester retina. The onset of CRB1 and CRB2 expression and other members of the CRB complex in the human fetal retina was found to be recapitulated in early versus late differentiated human induced-pluripotent stem cell (iPSC)-derived retinal organoids [129].

In the adult mammalian retina CRB1, CRB2, and CRB3 proteins localize at the subapical region in the Müller glial cells of mice, non-human primates and humans (Figure 7C–E). However, while CRB3 is present at the subapical region of photoreceptors in all three species, the expression patterns of CRB1 and CRB2 at the subapical region of photoreceptors differs between the three species (Figure 7C–E). CRB2 is present at the subapical region of mouse photoreceptors, whereas CRB1 is not (Figure 7C). In non-human primates both CRB1 and CRB2 are present in photoreceptors (Figure 7D). In cadaveric human retinas collected 2-days after death CRB1 is present at the subapical region whereas CRB2 is present at vesicles in the photoreceptor inner segments but at some distance from the subapical region (Figure 7E) [130,132,270,271]. Interestingly, all 3 CRB proteins in the adult human retina are detected in the photoreceptor inner segments at a distance from the outer limiting membrane [272]. Additionally, CRB3 is also detected in the inner retina of mice, non-human primates and humans [132,270,273]. All of the ultrastructural immuno-electron microscopy studies using anti-CRB on adult human cadaveric retina were carried out using two-days-old tissue samples, and might for that reason differ with the results obtained when using freshly collected non-human-primate retinas. In other studies, a negative correlation between protein abundance and post-mortem time has been found in the human retina [274,275].

In the adult mouse RPE, 1st and 2nd trimester human fetal RPE, and human iPSC-derived RPE CRB2 has been found to be expressed [129,276]. In the both human fetal and iPSC-derived RPE CRB2 immuno-EM labeling was found above adherens junctions and tight junctions in the apical membrane and microvilli [129].

## 9. CRB1 and Leber Congenital Amaurosis

Leber congenital amaurosis (LCA) is an early-onset disease leading to blindness from near birth with *CRB1* mutations accounting for 7–17% of cases and affecting approximately 10,000 patients worldwide [235,277,278,279]. Currently, there is no treatment available for *CRB1*-LCA patients, but proof-of-concept gene supplementation studies have shown functional rescue in *CRB1* retinitis pigmentosa (RP) mouse models which sets a ground work for proof-of-concept in *CRB1*-LCA-like mouse models [272]. *CRB1* gene mutations cause LCA8 with patients having severely attenuated or non-recordable ERG, abnormal pupillary reflex and nystagmus and their retinas have abnormal layering with reports of the retina being thickened, thinned or unchanged [280,281,282,283,284,285,286,287,288,289]. Mutations in CRB1 also lead to the development of RP with patients developing night blindness and a progressive loss of visual field due to rod degeneration [92]. *CRB1* mutations are associated with RP type 12 and are characterized by preservation of the para-arteriolar retinal pigment epithelium (PPRPE) and, due to macular involvement, a progressive loss of visual field. Disease onset can be from early childhood, however, some patients do not exhibit symptoms until after the first decade of life [290,291,292]. Additional CRB1 clinically relevant features include macular atrophy, keratoconus, Coats-like exudative vasculopathy, RP without PPRPE, pigmented paravenous chorioretinal atrophy, and nanophthalmos [288,293,294,295,296,297].

Human stem cell-derived retinal organoids which mimic retinal disease are an excellent tool for understanding disease mechanisms as well as a platform for testing therapeutic strategies. Other teams, including our own, are investigating pathobiology and disease mechanism of stem cell-derived retinal organoids from *CRB1* patients. We, have recently shown for the first time that three hiPSC lines (LUMC0116iCRB; LUMC0117iCRB; LUMC0128iCRB) from *CRB1* RP patients showed a phenotype similar as previously found in 3 month-old *Crb1^KO^* RP mice when differentiated into retinal organoids [129,130]. LUMC0116iCRB has c.3122T>C p.(Met1041Thr) homozygote missense mutations. LUMC0117iCRB has 2983G>T p.(Glu995*) and c.1892A>G p.(Tyr631Cys) mutations. LUMC0128iCRB has c.2843G>A p.(Cys948Tyr) and c.3122T>C p.(Met1041Thr) missense mutations. Mutations are mapped to their protein location in Figure 6C. Compared to control retinal organoids (Figure 8A), *CRB1* patient derived retinal organoids at DD180 showed disruptions at the OLM demonstrated by ectopic photoreceptor nuclei above the OLM and altered localization of CRB complex members at the OLM (Figure 8B,D). Furthermore, the CRB1 variant proteins localized to the subapical region above the adherens junctions, as in controls, but additionally showed a curved and broadened expression pattern (Figure 8C,E). Mislocalization of the CRB1 variant protein was also found mislocalized in the apical area of the NBL [129]. However, these studies are in their early stages with the underlying effects of the variant CRB1 proteins on protein-protein interactions and downstream cell signaling pathways still to be elucidated. Many rodent models of CRB retinal degeneration exist and have provided us with a deep insight into the pathobiology and mechanisms underlying CRB disease [130,144,232,235]. *CRB1*-LCA-like models have further built on our previous data [132,298,299].

One of the main working hypothesis that we draw from our four *CRB1*-LCA-like models is that CRB2 protein levels may be lower or that a less functional variant is of CRB2 is expressed in *CRB1* LCA patients compared to less severe *CRB1* retinal diseases [132,298,299]. Clinical reports of *CRB1* LCA patients have shown that degeneration can affect all quadrants of the retina while other reports show restriction to the inferior retina [285,288,300,301]. This is highly suggestive of the presence of a modifying factor. Variants of *CRB2* have been associated with retinal aberrations and more recently a missense mutation of *CRB2* has been found to cause retinitis pigmentosa [134,302]. Additionally, CRB2 is present in the fetal human retina in the first-trimester, whereas CRB1 expression starts from the second trimester [129]. In the mouse, both CRB1 and CRB2 are found in the early embryonic retina [232,242,269]. Therefore, missense variants of *CRB2* in humans are likely to have small but important effects on retinal diseases. A number of transcript variants have been reported for CRB1, and new novel isoforms of CRB1 are reported in the mouse and human retina and may also lead to the phenotypic variability in *CRB1* patients [144,303]. Currently there has been no link between *CRB3* mutations and retinal disease.

The four *CRB1*-LCA-like models had both alleles of *Crb2* disrupted in either retinal progenitor cells (*ΔRPC*), immature photoreceptors (*ΔimPRC*), or Müller glial cells (ΔMG) on genetic backgrounds with either reduced levels of (*Crb1^KO/WT^Crb2^ΔRPC^*) or complete knockout of *Crb1* (*Crb1^KO^Crb2^ΔRPC^*, *Crb1^KO^Crb2^ΔimPRC^*, *Crb1^KO^Crb2^ΔMG^*). All of these models had abnormally layered and transiently thickened retina, disruptions of the outer limiting membrane and ectopic localization of mitotic progenitors, cycling cells, and immature photoreceptors. The thickened retinas observed were in part due to the ectopic birth or displacement of early progenitors which we found increased in the *Crb1^KO^Crb2^ΔRPC^* and *Crb1^KO^Crb2^ΔimPRC^* mouse retinas, but not in the *Crb1^KO^Crb2^ΔMG^*. This led to adult retinas in which we detected ectopic cells in the ganglion cell layer either sporadically (*Crb1^KO^Crb2^ΔRPC^*), at the peripheral retina (*Crb1^KO^Crb2^ΔMG^*), or within most of the retina (*Crb1^KO^Crb2^ΔimPRC^*), or throughout the retina (*Crb1^KO^Crb2^ΔRPC^*). We hypothesize that *CRB1* LCA patients which exhibit a thickened retina and abnormal layering do so due to similar mechanisms as found in our *CRB1*-LCA-like mouse models, displacement or the ectopic birth of progenitor cells, cycling cells, and immature photoreceptor cells [132,285,298,299]. Additionally, the reported thinned or unchanged retinal thickness of *CRB1* LCA patients is likely due to measurements made when significant retinal degeneration had already occurred. In our *CRB1*-LCA-like mouse models we detected transient changes in retinal thickness [288,289].

While in the *Crb1^KO/WT^Crb2^ΔRPC^* and *Crb1^KO^Crb2^ΔRPC^* mouse models no differences in retinal degenerations were reported between superior/inferior or central/peripheral retina, the *Crb1^KO^Crb2^∆imPRC^* and *Crb1^KO^Crb2^ΔMG^* retina showed superior/inferior or central/peripheral phenotypes, respectively [132,298,299]. Interestingly, a new *Crb1Crb2* double knockout mouse model which disrupted both alleles of *Crb2* in rods (Δrods) on a genetic background lacking *Crb1* (*Crb1^KO^Crb2^Δrods^*) does not have an LCA-like but RP-like phenotype. The *Crb1^KO^Crb2^Δrods^* retinas have a phenotype that mainly affects the peripheral and central superior retina [304]. These differences may be attributed to opposing gradients of mouse CRB1 and CRB2 at the subapical region between the superior and inferior retina as well as the contribution of CRB1 and CRB2 to either photoreceptors (CRB2) or Müller glial cells (CRB1 and CRB2) [130,270,271]. Another common feature of the *CRB1*-LCA-like mouse models is the early formation of retinal rosettes [132,298,299]. A comparison of the *Crb1Crb2* double knockout mouse models can be found in Table 1.

Rosette formation has been extensively described in many retinal conditions including retinitis pigmentosa, diabetic retinopathy, and retinoblastoma [305,306,307]. The formation of rosettes has been attributed to the disruption of the OLM both chemically and genetically [130,231,233,266,308,309,310,311,312]. A defining hallmark of CRB mouse models is the formation of photoreceptor rosettes which may be concurrent with the loss of polarity in a CRB dependent manner. The more severe CRB models have a low level of total CRB, less stable AJs and thus an earlier phenotype onset and rosette formation. The less severe CRB models have a higher level of total CRB, more stable AJs and thus a later phenotype onset and rosette formation [144]. Rosette formation is preceded by aberrant localization of retinal cells into the subretinal space at foci where loss of adhesion is found. In the developing retina, this is usually seen as “volcanic-like” cell eruptions, while in the mature retina this is seen as a loss of complete rows of photoreceptors [130,131,132,232,270,298,299]. The *Crb1^KO^Crb2^ΔMG^* retina has a peripheral to central degenerative phenotype. At P1 in the *Crb1^KO^Crb2^ΔMG^,* retina rosettes can be detected peripherally while only protrusions into the subretinal space are found in the central retina. By P14 the peripheral rosettes are gone due to advancement of the phenotype, but rosettes can now be found in the central retina [132]. Differences in the number of aberrant cells in early- versus late-disrupted OLM phenotypes likely arise from both the extent of OLM disruption as well as the developmental stage of the retina. Earlier cells are less mature/competent and are still undergoing division and migration changes. Rosette formation is likely independent of any increased cell proliferation seen in early-onset RP-like and LCA-like CRB models as rosettes are also present in later onset CRB, CRB-related, and non-CRB mouse models that do not exhibit changes in proliferation [130,131,132,232,298,299,310,311,313].

Although the mechanism by which rosettes are formed in retinal disease is not fully alluded to, it has been related to changes in the extracellular matrix, adhesion molecules, and the cytoskeleton, all of which can be affected by an imbalance between apical and basal polarity domains [314,315,316,317]. Loss of CRB in mutant mouse models may not be uniform due to opposing gradients of CRB1 and CRB2 proteins in the superior versus inferior retina as well as the non-uniform localization of CRB between photoreceptors (CRB2) and Müller glial cells (CRB1 and CRB2), leading to imbalance of CRB levels at the OLM in the mutant models [130,270,271]. Differences in adjacent apical levels of CRB are well studied as this is part of the process for tube invagination, similar mechanisms in opposing CRB levels between adjacent cells may lead to rosette formation, which are in fact invaginations of the apical retinal surface [318,319,320].

## 10. Gene Augmentation for Hereditary Retinopathies

Gene augmentation, via recombinant adeno-associated virus (rAAV) delivery, is currently the “go to” therapeutic strategy for targeting hereditary retinal diseases. Luxturna (voretigene neparvovec-rzyl) has led the way for eye gene therapeutics by becoming the first FDA- and EMA-approved AAV gene therapy for patients with biallelic *RPE65* gene mutations [321,322]. As such a large number of AAV mediated gene therapeutics are currently going under clinical trials focused on the treatment of achromatopsia, autosomal recessive retinitis pigmentosa, choroideremia, Leber hereditary optic neuropathy, retinitis pigmentosa, X-linked retinitis pigmentosa, X-linked Retinoschisis, and wet age-related macular degeneration (summarized in Table 1, Alves and Wijnholds 2018 [323]). The clinical success of rAAV mediated eye gene therapeutics is in part due to its low toxicity; the small amount of rAAV required to infect the retinal pigment epithelium or retina; the surgical accessibility of the eye; the large number of non-invasive techniques for monitoring disease progression, such as ERG, scanning laser ophthalmoscopy (SLO) and spectral domain optical coherence tomography (SD-OCT); and the immune-privileged status of the eye, having a good safety profile with low immunogenicity. Additionally, rAAV vectors can transduce both dividing but also non-dividing cells such as photoreceptors and display varied cell tropisms due to a plethora of capsid variants [323,324].

Wild type AAVs are small non-enveloped single stranded DNA viruses that belong to the parvovirus family in the genus Dependovirus and as such require assistance for replication. Cell surface receptors such as heparin sulfate and sialic acid mediate AAV endocytosis. AAV is then processed through the cytosol too the nucleus where it is uncoated and processed into nuclear episomal structures. The AAV genome includes three open reading frames which express the Replication (*rep*), Capsid (*cap*) and assembly activating protein (*aap*) genes. Two T-shaped 145 nucleotide-long inverted terminal repeats (ITRs) flank the genome. The ITRs and the four proteins encoded by the *rep* gene, Rep78, Rep68, Rep52, and Rep40, are needed for genome replication and packaging. Virion proteins (VP1), VP2, and VP3 from *cap* gene transcripts form an icosahedral symmetry shell ~26 nm in diameter defined by 60 subunits in a molar ratio of 1:1:10 (VP1:VP2:VP3). The assembly of the virions is promoted by the scaffolding function of AAP. However, in rAAV the *rep* and *cap* genes and the element required for site-specific integration into the genomic locus *AAVS1* are deleted. This means for replication a helper plasmid containing the *rep* and *cap* genes along with helper genes from adenovirus (*E4*, *E2a,* and *VA*) must be supplied [324].

Development of a *CRB*-based gene therapy approach was particularly challenging due to the cDNA size of 4.2 kb for human *CRB1*. With the addition of promoter, polyadenylation, and ITR sequences this is at the very edge of the approximate 4.7 kb packaging space of rAAV. Additionally, human *CRB1* cDNA required codon optimization to achieve sufficient levels of expression. However, alternative strategies employing the use of the 3.85 kb human *CRB2* cDNA and the development of minimal promoters for the expression of CRB proteins in photoreceptors and Müller glial cells has led to proof-of-concept for a *CRB*-based gene therapeutic [272,325]. These proof-of-concept pre-clinical studies found both morphological and functional rescue in two *CRB1*-RP-like mouse models when using an AAV-*CRB2*-based gene therapy vector. We found that expression of *CRB1* was deleterious in *CRB1* RP-like mouse models but not in wild-type mice. The *CRB2*-based gene therapy used a combination of AAV9 and the ubiquitous CMV promoter to target both photoreceptors (cone and rods) and Müller glial cells to achieve rescue. No rescue was achieved if either photoreceptors or Müller glial cells were targeted independently. We hypothesize that physiological relevant levels of CRB proteins are required at the subapical regions in neighboring photoreceptors and Müller glial cells [272]. There are a large number of transcript variants for CRB1 [144]. The Kay lab has identified a number of novel CRB1 isoforms in both mouse and human retina [303].

Cultured human iPSC-derived retinal organoids recapitulate well the human fetal retina, and might therefore be good models to test gene therapy vectors [143]. Recently we used transgene expression assays in human iPSC-derived retinal organoids and adult cadaveric human retinal explants to assess the tropism of three AAV serotypes: AAV9, AAV5, and ShH10Y using the ubiquitous CMV promoter. We found a preference of AAV5 and ShH10Y445F over AAV9 to infect Müller glial cells in hiPSC-derived retinal organoids. We observed as well a higher efficacy of AAV5 than ShH10Y445F or AAV9 serotypes for infection of photoreceptors and Müller glial cells in cultured human donor retinal explants. Together, our data indicate that AAV5 serotype in combination with the CMV promoter may be a viable strategy to express the *CRB* gene in human photoreceptors (rods and cones) and Müller glial cells [129]. An additional clinically relevant finding we discovered was the higher efficacy of AAV5-CMV-*GFP*, ShH10Y445F-CMV-*GFP*, and AAV9-CMV-*GFP* to express in Müller glial cells than photoreceptors in human retinal explants lacking photoreceptor segments. This indicated (1) that in the absence of photoreceptor segments there is an increased bioavailability of AAV vectors, allowing targeting of less-abundant/preferred receptors for AAV uptake; (2) that there may be a common mechanism of active AAV uptake to photoreceptors through there segments, e.g., putative sites of receptor-dependent or -independent clathrin- and caveolae-mediated endocytosis; (3) that intact photoreceptor segments allow for efficient AAV5 gene therapy vector transduction of human photoreceptors, whereas loss of intact photoreceptor segments allows for efficient AAV5 gene therapy vector transduction of Müller glial cells [129].

## 11. Concluding Remarks

Human stem cell-derived retinal organoids faithfully recapture in part many of the facets of the human fetal retina, including retinal cell type diversity, morphological cues, synaptic wiring, and light sensitivity. Improved retinal organoid culturing methods using bioreactors or microfluidic organ-on-a-chip technology, which also allows for tight control of the physiological microenvironment, bring us a further step towards an in vivo like retina [167,215,326]. Furthermore, retinal organoid models derived from patients may be used in conjunction with two-photon imaging and light sheet microscopy as well as tissue clearing methods such as DISCO and PACT [47,169,327,328,329]. This will allow unparalleled analysis of live and fixed cellular events including spatiotemporal process such as proliferation, differentiation and migration. Additionally, new methods such as ferrofluid droplets as mechanical actuators allow analysis of the mechanics of 3D developing tissues. This tool in combination with optogenetics or calcium imaging would provide insight into how neuronal and mechanical responses may influence each other in retinal development and disease [330,331]. Together, these are all valuable tools to further evaluate the mechanisms by which the misregulation of the apical CRB and PAR complexes affects retinogenesis leading to the severe retinal degeneration seen in LCA patients. Additionally, *CRB*-based gene augmentation is a viable option for *CRB1*-related retinitis pigmentosa and needs to be further evaluated for *CRB1*-related LCA.

## Figures and Tables

**Figure 1 genes-10-00987-f001:**
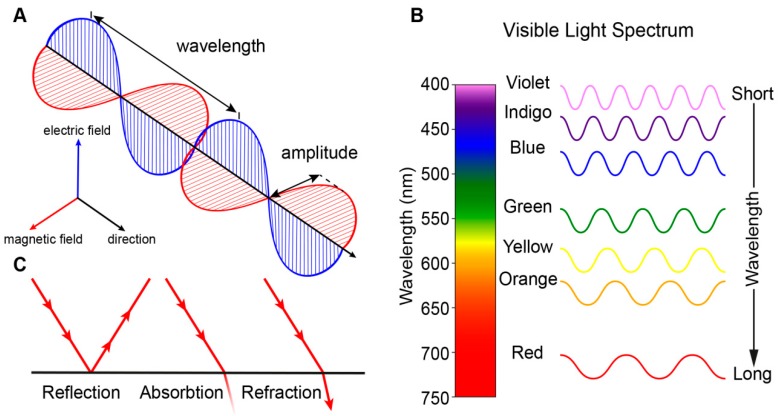
Transmission of light. (**A**) Light is electromagnetic radiation that travels as waves consisting of perpendicular oscillating electric and magnetic fields. (**B**) Visible light is a narrow group of wavelengths between approximately 400 nm (short wavelength) and 760 nm (long wavelength) which we interpret as a spectrum of different colors. Wavelengths outside this range are not visible to humans. (**C**) Light can be reflected, absorbed and refracted.

**Figure 2 genes-10-00987-f002:**
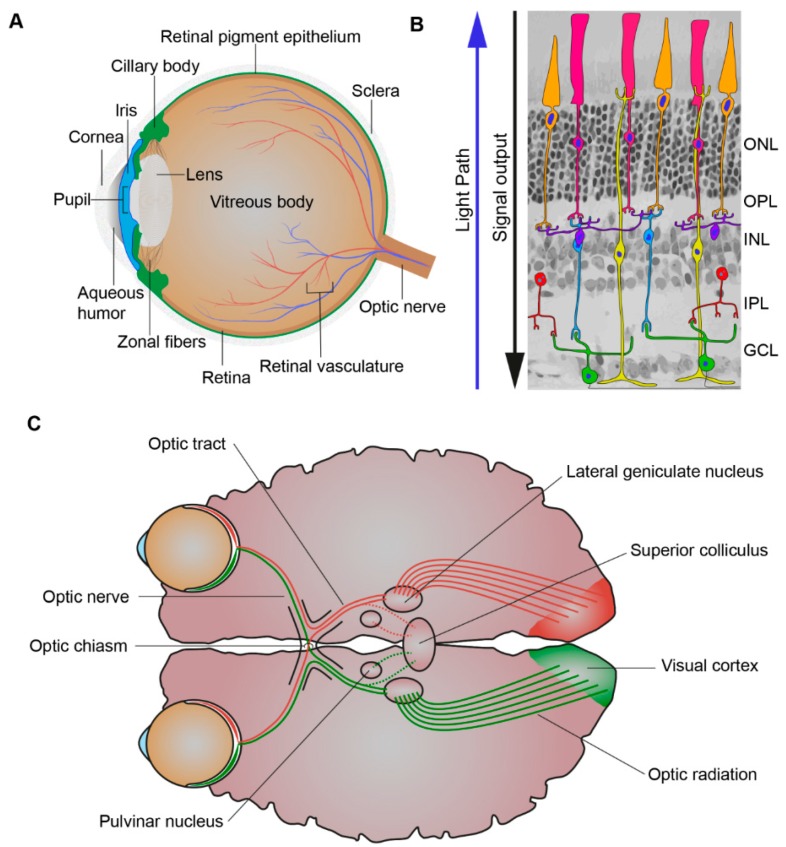
Processing of light. (**A**) Schematic picture of the eye. The eye is comprised of the aqueous humor, ciliary body, cornea, iris, lens, optic nerve, pupil, retina, retinal pigment epithelium, retinal vasculature, sclera, vitreous body, and zonal fibers. When light first enters the eye, it is refracted by the cornea through the pupil, whose size is controlled by the iris. The iris, the colored part of the eye, controls the amount of light entering the eye while the lens focuses the light through the vitreous humor and on to the proximal surface of the retina. (**B**) Schematic picture of the retina. The retina is composed of seven cell types: amacrine cells (red), bipolar cells (blue), cones (orange), ganglion cells (green), horizontal cells (purple), Müller glial cells (yellow), and rods (pink). When light first enters the retina, it goes through the ganglion cell layer (GCL), then the inner plexiform layer (IPL), inner nuclear layer (INL), outer plexiform layer (OPL), and outer nuclear layer (ONL). As light is passing through the retina it is absorbed by its light responsive cells: rod and cone photoreceptors and the intrinsically-photosensitive retinal ganglion cells (ipRGCs). This creates electrophysiological signals that then are further propagated to the inner retina and can be propagated through many different cell to cell pathways to the ganglion cells. (**C**) Schematic picture of the visual pathway. The axons of the retinal ganglion cells exit the eyes as bundles, the optic nerve, and extend to the optic chiasm were the nasal axons of each eye crossover and combine with the contralateral eyes temporal axons and subsequently via the optic tract travel to the lateral geniculate nucleus (LGN) and superior colliculus (SC). The LGN, SC, and pulvinar nucleus are all involved in the process of relaying and refining visual information to the visual cortex. Visual information is relayed to the visual cortex via optic radiations which extend from the LGN.

**Figure 3 genes-10-00987-f003:**
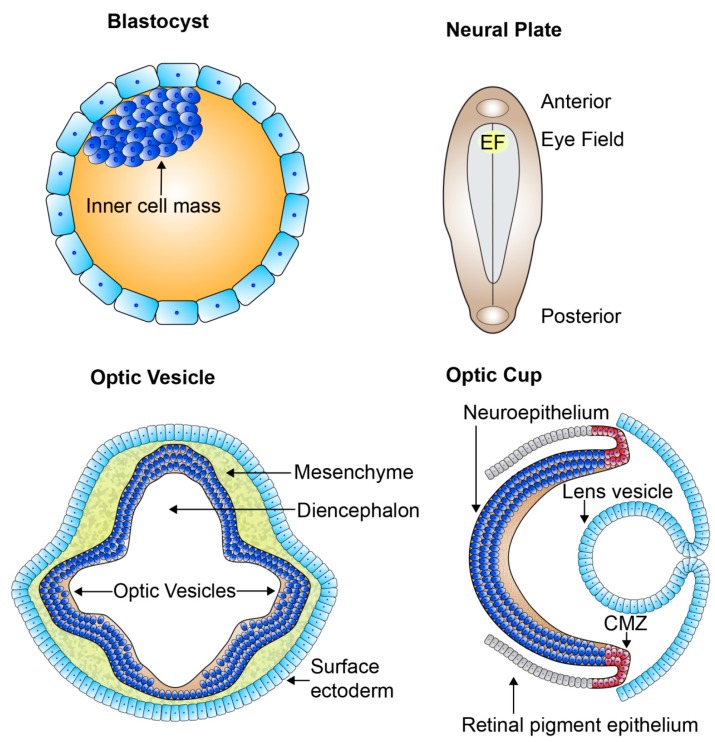
The organization of the developing retina. Schematic picture of early retinal development. From the blastocyst which contains the pluripotent cell mass gastrulation and neurulation occur forming the neural plate. The eye field specifies at the medial region of the anterior neural plate and contains all the progenitors which go on to form all the neural-derived cell types and structures of the eye. Bilateral optic sulci develop from the eye field forming the optic vesicles which extend towards the surface ectoderm. The optic vesicles invaginate forming the two-layered optic cups and the lens vesicle forms and sits behind the surface ectoderm. The outer layer of the optic cup remains as a single cuboidal layer becoming the retinal pigment epithelium. The single inner layer of the optic cup proliferates and differentiates forming the multilayered neural retina. EF: eye field; CMZ: ciliary marginal zone.

**Figure 4 genes-10-00987-f004:**
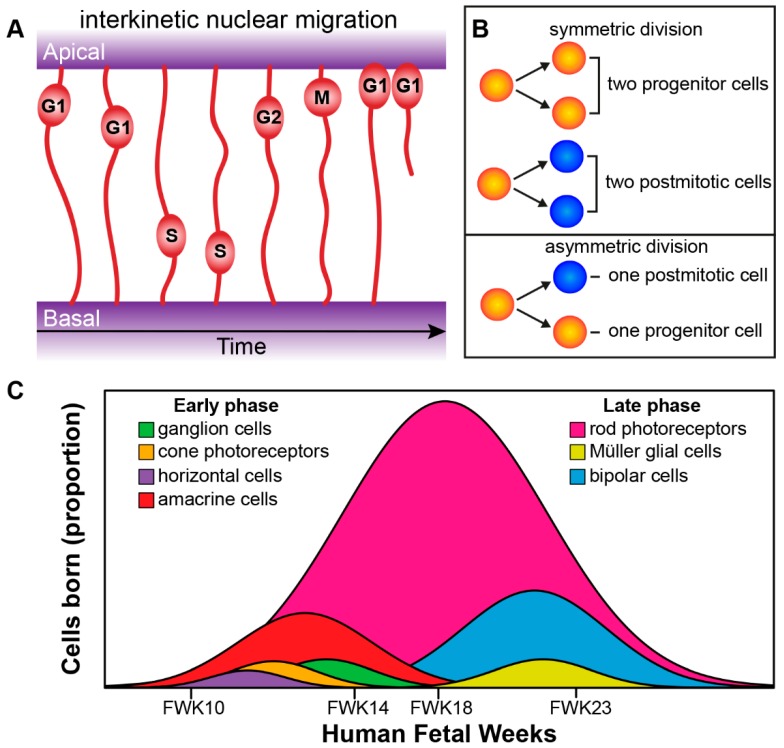
Retinogenesis. (**A**) Radial progenitor cells undergoing interkinetic nuclear migration during cell cycle phases G1, S, G2, and M. The mitosis (M) phase takes place at the apical side, whereas the DNA synthesis (S) phase takes place more basally. (**B**) Symmetric versus asymmetric cell division. (**C**) Genesis of retinal cells born during the development of the human eye can be divided into an early phase (ganglion cells, cone photoreceptors, horizontal cells, and amacrine cells) and an overlapping late phase (rod photoreceptors, Müller glial cells, and bipolar cells; see Aldiri et al. 2017 [40]). FWK—fetal week.

**Figure 5 genes-10-00987-f005:**
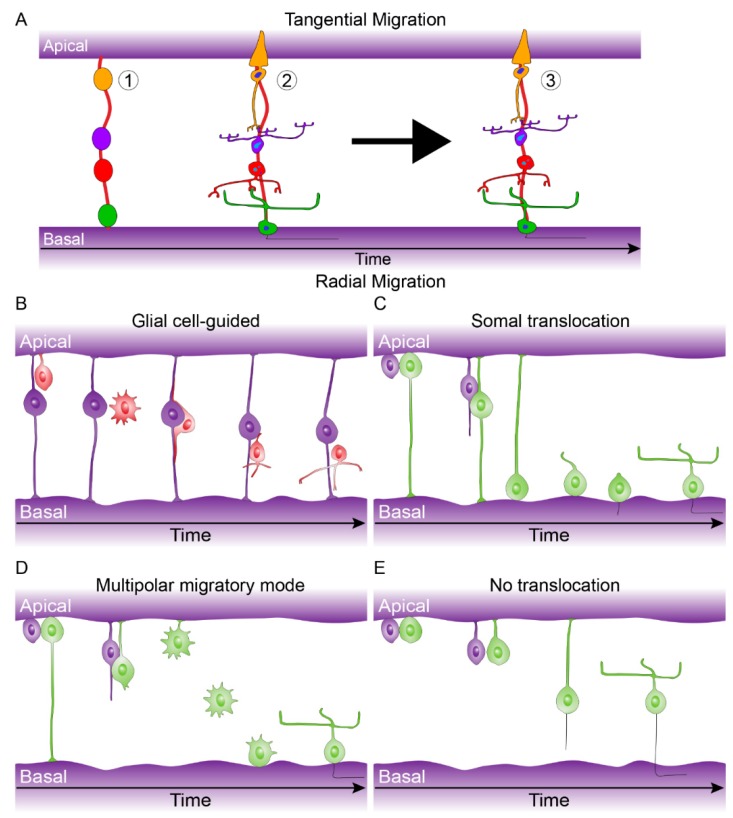
Tangential and Radial migration. (**A**) Tangential migration can be described in three steps: (1) Early born cell type progenitors localize to their correct laminar position (Cones: orange, Bipolar cells: purple, amacrine cells: red, ganglion cells: green), (2) they undergo morphological differentiation, (3) tangential migration coincides with morphological differentiation allowing subsets of early born cell types to move a short distance within their laminar position (see Reese et al. 1999 [43]). (**B**) Glial cell-guided, apically born neurons become initially detached and subsequently attach to radial glial progenitor cells. They then migrate along the radial glial progenitor cells to the target laminar location where they fully integrate. (**C**) Somal translocation, apically born nuclei can move along there inherited basally attached process from. Once they move to their final laminar location they fully integrate (This process can also occur with only apically inherited processes). (**D**) Multipolar migratory mode, in rare case apically born neurons can loses both apical and basal attachments but can move to their final laminar position and integrate due to a multipolar mode. (**E**) No translocation, inefficient migration due to retention of the apical or basal process and slow release of opposing process. For further details see Icha et al. 2016 and Amini et al. 2018 [45,47].

**Figure 6 genes-10-00987-f006:**
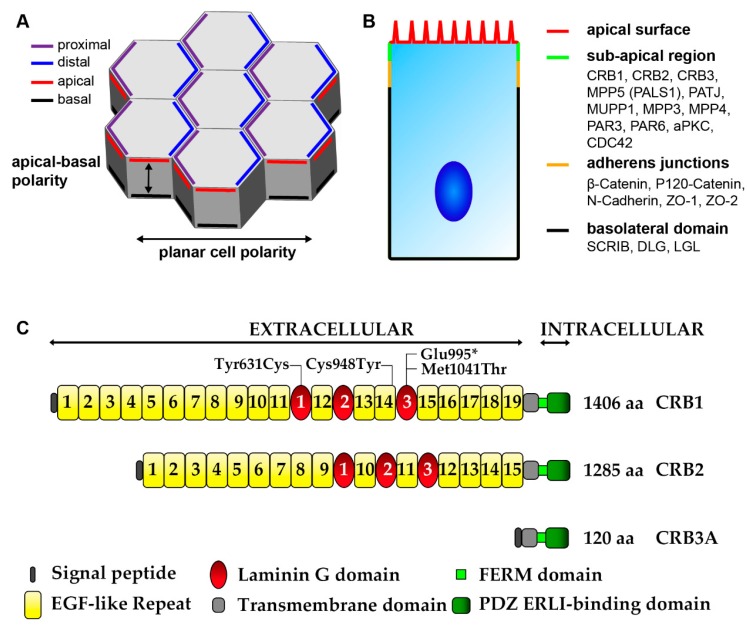
Schematic picture of Epithelial polarity. (**A**) Epithelial cell polarity consist of apical-cell polarity and orthogonal to this axis planar cell polarity (PCP). PCP is the collective alignment of cell polarity across the tissue plane and involves asymmetric segregation of proximal (purple) and distal (blue) PCP components. Apical-basal polarity involves the antagonistic, functional, and spatial segregation of apical (red) and basal (black) components. (**B**) In the retina the CRB and PAR complexes are located at the sub-apical region (green), below the apical surface (red) and adjacent to the adherens junctions (yellow). Scribbled (SCRIB), discs large (DLG), and lethal giant larvae (LGL) form a basolateral domain (black) extending below the adherens junctions. (**C**) Schematic overview of the prototypic CRB1, CRB2, and CRB3 proteins. Mutations from the three iPSC CRB1 RP patients lines recently derived to retinal organoids are mapped to their protein location in CRB1. Adapted from Quinn et al. 2017 [144].

**Figure 7 genes-10-00987-f007:**
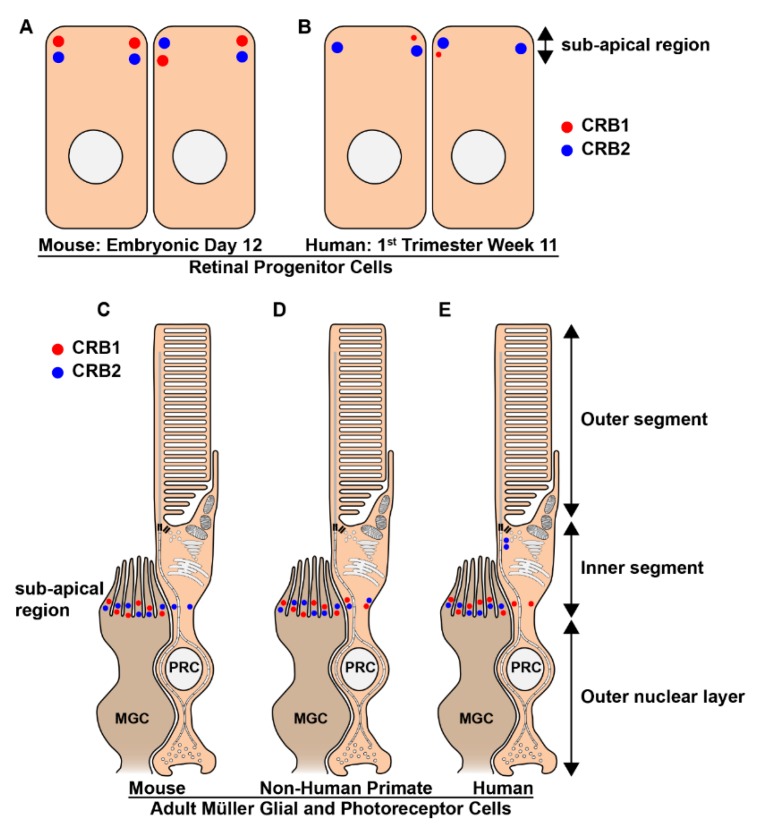
Schematic picture of the localization of the CRB proteins in mammalian adult retinal Müller glial and photoreceptor cells. (**A**–**E**) CRB1 (red) and CRB2 (blue) proteins localize at the subapical region in mouse (**A**) and human (**B**) retinal progenitors cells and adult Müller glial cells of mice (**C**), non-human primates (**D**) and humans (**E**). However, the expression of CRB1 in the early developing human retina is low and sporadic. Additionally, the expression patterns of CRB1 and CRB2 at the subapical region of adult photoreceptors differs between the three species. CRB2 is present at the subapical region of mouse photoreceptors, whereas CRB1 is not (**C**). In non-human primates both CRB1 and CRB2 are present in photoreceptors (**D**). In cadaveric human retinas collected 2-days after death CRB1 is present at the subapical region whereas CRB2 is present in the photoreceptor inner segments but at some distance from the subapical region (**E**). PRC: Photoreceptor; MGC: Müller glial cells.

**Figure 8 genes-10-00987-f008:**
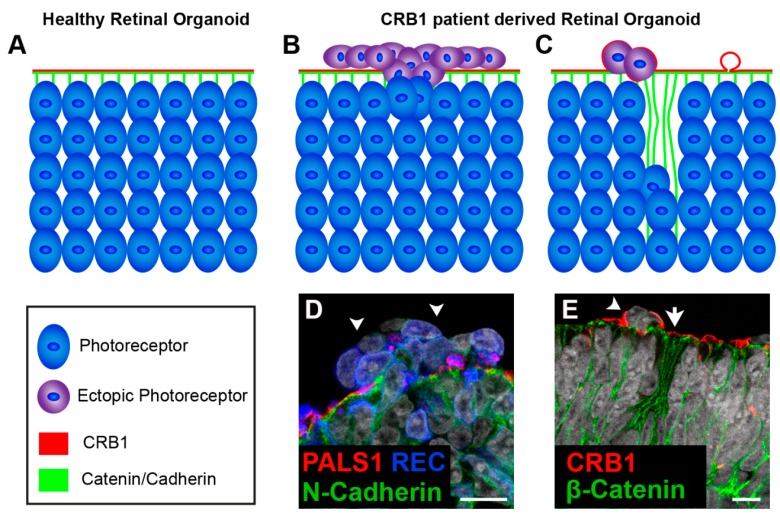
Disruptions at the outer limiting membrane in retinal organoids from *CRB1* retinitis pigmentosa (RP) patients. Schematic depiction of healthy (**A**) and *CRB1* patient (**B**,**C**) derived retinal organoids. (**A**) In healthy organoids CRB1 is located at the subapical region adjacent to the adherens junctions. (**B**,**D**) In retinal organoids from *CRB1* retinitis pigmentosa patient (LUMC0116iCRB) disruptions of the outer limiting membrane (OLM) and ectopic photoreceptors have been found. (**C**,**E**) Additionally, budding of the apical membrane and mislocalization of the CRB1 variant in the NBL also occurs. Scale bars (**D**,**E**), 20 µm. Panels (**C**,**D**) are modified from [178] under a Creative Commons license.

**Table 1 genes-10-00987-t001:** Comparison of Crb1Crb2 double knockout mouse models.

	*Crb1^KO^Crb2^ΔRPC^*	*Crb1^KO^Crb2^ΔimPRC^*	*Crb1^KO/WT^Crb2^ΔRPC^*	*Crb1^KO^Crb2^ΔMG^*	*Crb1^KO^Crb2^Δrods^*
**Severity**	++++	+++	++	++	+
**CRB1 + CRB2 in MGC/PRC/RPC**	0/0/0	60/0/60	20/0/20	0/100/60	60/03/60
**Cre Mosaicism**	50%	95%	50%	95%	99–100%
**Phenotype onset**	E13 whole retina	E15 whole retina	E15 periphery	E17 periphery	3M periphery
**OLM disruption**	++++	++++	++	+++ centrally ++++ at periphery	+ whole retina
**Abnormal retinal lamination**	YES	YES	YES	YES	NO
**Ectopic localization retinal cells**	YES of all neurons	YES of PRC and RPC	YES of PRC, BC, AC and GC	YES of PRC, MGC, BC, AC and RPC	YES of PRC
**Transiently thickened retina**	YES	YES	YES	YES	NO
**Intermingling of nuclei of the ONL and INL**	YES	YES	YES But moderately	YES at peripheral retina	YES Sporadic
**Ectopic retinal cells in GCL**	++++	+++	(+) sporadic ectopic cells	+ periphery	NO
ERG max b-waves 1M/3M	25%/0%	*/*	25%/10%	50%/0%	100%/90%
**IS/OS at P14**	NO	NO	NO	NO	YES
**ONL at P14**	NO	NO	YES	YES but not at periphery	YES
**OPL at P14**	NO	NO	At foci	YES but not at periphery	YES
**INL at P14**	NO	NO	YES	YES	YES
**IPL at P14**	YES	YES	YES	YES	YES
**GCL at P14**	Thickened in all of retina	Thickened in majority of retina	Thickened in part of retina	Thickened at peripheral retina	YES
**Increase number of RPC**	YES	YES	YES	NO	NA
**Severity retinal phenotype**	Throughout retina	Superior > inferior	Peripheral > central	Peripheral > central	Peripheral and Central Superior
**Neovascularization**	YES	YES	YES	YES	NA
**Photoreceptors lost at 3M**	YES (some sparse nuclei)	NA	thin ONL	YES (some sparse nuclei)	Minimal
**Retinal overgrowth by late-born cells**	YES	NO	NO	NO	NA
**Apoptosis increased**	YES	YES	YES	NA	NA

MGC—Müller glial cell; PRC—rod and cone photoreceptors; RPC—retinal progenitor cell; E—embryonic day; M—month; OLM—outer limiting membrane; BC—bipolar cell; AC—amacrine cell; GC—ganglion cell; ONL—outer nuclear layer; INL—inner nuclear layer; GCL—ganglion cell layer; ERG—electroretinography; *, mice not analyzed by ERG due to severe hydrocephalous; IS—inner segment; OS—outer segment; OPL—outer plexiform layer; IPL—inner plexiform layer; NA—not analyzed. Percentages distribution: CRB1 estimated 40% and CRB2 60% weight in MGC and RPC. CRB2 estimated 100% weight in PRC. ^#^ Rods estimated at 97% and cones at 3%. Adapted from [132].

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
