# Peer review of "Retinogenesis of the Human Fetal Retina: An Apical Polarity Perspective"

_genes, 2019, doi:10.3390/genes10120987_

Round 1

Reviewer 1 Report

The review by Quinn and Wijnholds is a well written text, with very illustrative figures.

Nonetheless, it is a bit surprising that such a comprehensive review on the formation, development and maintenance of the retinal structure in mammals is only centered in the cell biology and physiology perspective, with hardly any mention at all to the genetics behind all the developmental and differentiation control, particularly since this work is to be published in a journal entitled "Genes". Although I do not have many comments on what has been included, I am a bit concerned by what it is missing, for instance, the lack of references to retinal genetics, more even so since most rare diseases (including Leber Congenital Amaurosis, to which the authors make most references) are caused by mutations in genes relevant to the formation and maintenance of retinal structures and physiological function.

I understand that the main focus of the review is the contribution of the CRB genes in the apical polarity of the retina and the authors wish to strengthen that: i) retinal organoids recapitulate most of the cellular structure and molecular pathways of the in vivo retinas, and ii) CRB2genetic variants contribute as genetic modifiers to the phenotype of CRB1mutations . However, I still believe that the general reader will greatly benefit from a genetics perspective, even if brief, without much change to the present text, and based only on additions to the text.

If then the number of words is too large, I would suggest to cut back on the explanation on AAV viral vectors (text between lines 672 to 685 could be shortened) that is not required for text comprehension.

These are my suggestions:

1) I would suggest adding a whole paragraph or even better, a small section, between sections 2 (Retinogenesis) and 3 (Morphological and Molecular Recapitulation of the Human Fetal Retina) to add genetic relevant information to the main arguments of the. There are many genetic, transcriptomic, and single-cell transcriptomics on the mouse and human fetal retinas, as well as on retinal organoids, which will further support the two main points of the authors (among others, work by the groups of Swaroop, Liu...).

2) Since the authors wish to stress the contribution of CRB2 variants to the CRB1 mutant phenotype in humans (lines 584 onwards), I would suggest to look for genetic evidences in human population, or at least mention whether their hypothesis is supported by data in humans, not only by genetic manipulation in organoids or mouse models. On the other hand, if CRB2 variants modify CRB1 mutations, what about CRB3 variants? There are neither comments nor hypotheses on the redundancy of function, strength of genetic interactions between all the CRB genes in human disease or in physiology. I would suggest to add some sentences to strengthen the authors' arguments.

3) It is difficult to understand why if the gene mutated in humans is CRB1in humans, why a CRB2-based gene therapy would be more effective, without even considering to deliver a mini-CRB1gene (as it is now being explored for dystrophin in DMD and CEP290in LCA). This strategy should be at least mentioned.

4) There are several minor typos, for instance, in line 44 and 85, I would suggest to substitute the relative pronoun "which" by "whose."

Author Response

Reviewer 1

The review by Quinn and Wijnholds is a well written text, with very illustrative figures.

Nonetheless, it is a bit surprising that such a comprehensive review on the formation, development and maintenance of the retinal structure in mammals is only centered in the cell biology and physiology perspective, with hardly any mention at all to the genetics behind all the developmental and differentiation control, particularly since this work is to be published in a journal entitled "Genes". Although I do not have many comments on what has been included, I am a bit concerned by what it is missing, for instance, the lack of references to retinal genetics, more even so since most rare diseases (including Leber Congenital Amaurosis, to which the authors make most references) are caused by mutations in genes relevant to the formation and maintenance of retinal structures and physiological function. I understand that the main focus of the review is the contribution of the CRB genes in the apical polarity of the retina and the authors wish to strengthen that: i) retinal organoids recapitulate most of the cellular structure and molecular pathways of the in vivo retinas, and ii) CRB2 genetic variants contribute as genetic modifiers to the phenotype of CRB1 mutations. However, I still believe that the general reader will greatly benefit from a genetics perspective, even if brief, without much change to the present text, and based only on additions to the text. If then the number of words is too large, I would suggest to cut back on the explanation on AAV viral vectors (text between lines 672 to 685 could be shortened) that is not required for text comprehension.

These are my suggestions:

1) I would suggest adding a whole paragraph or even better, a small section, between sections 2 (Retinogenesis) and 3 (Morphological and Molecular Recapitulation of the Human Fetal Retina) to add genetic relevant information to the main arguments of the. There are many genetic, transcriptomic, and single-cell transcriptomics on the mouse and human fetal retinas, as well as on retinal organoids, which will further support the two main points of the authors (among others, work by the groups of Swaroop, Liu...).

Added additional section as suggested.

2) Since the authors wish to stress the contribution of CRB2 variants to the CRB1 mutant phenotype in humans (lines 584 onwards), I would suggest to look for genetic evidences in human population, or at least mention whether their hypothesis is supported by data in humans, not only by genetic manipulation in organoids or mouse models. On the other hand, if CRB2 variants modify CRB1 mutations, what about CRB3 variants? There are neither comments nor hypotheses on the redundancy of function, strength of genetic interactions between all the CRB genes in human disease or in physiology. I would suggest to add some sentences to strengthen the authors' arguments.

Added additional information on the new RP causing missense mutation found for CRB2 in humans.

3) It is difficult to understand why if the gene mutated in humans is CRB1 in humans, why a CRB2-based gene therapy would be more effective, without even considering to deliver a mini-CRB1gene (as it is now being explored for dystrophin in DMD and CEP290in LCA). This strategy should be at least mentioned.

Add additional information to Section 10. Gene Augmentation for Hereditary Retinopathies.

4) There are several minor typos, for instance, in line 44 and 85, I would suggest to substitute the relative pronoun "which" by "whose."

Changed as suggested.

Reviewer 2 Report

In this manuscript, after first presenting broad accounts of visual system structure and function and retinal development and describing a retinal organoid model system that has emerged relatively recently, its main focus, the Crb protein complex and its interacting partners are introduced. It then summarizes Crb1 and 2 protein distribution, primarily in the OLM of Muller glia and photoreceptors of mice, non-human primates and humans.  Subsequent sections describe findings from retinal organoids derived from human patient iPSCs carrying missense mutations of Crb1 and consider the pathogenic mechanisms of Crb1/2 based on studies of mouse genetic models.  To support their contention that Crb2 is a major contributing factor to the variability in onset and severity of Crb1 LCA and RP pathogenesis, the authors then summarize their previous work using various Cre drivers to delete Crb2 in the Crb1 null background. They conclude by addressing the utility of gene therapy, which is a prominent and promising therapeutic approach to retinal dystrophy of genetic origin, in Crb1 patients. Because the authors’ lab has accumulated experimental data and published extensively in this field of research, the review’s focus is primarily on the authors’ own findings. Overall, the review includes a wide array of information about interesting disease-relevant genes, genetic and in vitro models, and therapeutic approaches. The manuscript can be improved by better organization of some paragraphs and clarification of some of the following issues.  

Major points:

In Figure 3, what is the significance that G1 cells after mitosis (right side of cells in the mitosis) do not have apical endfeet? Does it expressly indicate that cells are out of the cell cycle (G0) or that cells transiently lose their apical endfeet during interkinetic migration? The authors should clarify this point. The first section about retinogenesis primarily focuses on progenitor cell division and fate acquisition, which should be separated from migration of postmitotic retinal cells. Therefore, the tangential migration portion of Figure 4 may be combined with the radial migration shown in Figure 5; this change would improve the organization of these Figures and the section. Section 6, titled “The Apical CRB and PAR Complexes,” would be improved significantly by reorganizing and clarifying whether the findings apply to other tissue or epithelial cell culture or to the retinal cells. It would be best to first describe the cell biological significance of the interaction between Crbs and binding partners and then describe the relevance to retinal biology and retinal dystrophy. Some animal models, such as Nr2e3rd7 and Nrl-/- mice, do not have sufficient information regarding their genetic cause or phenotype, which makes it difficult to understand the meaning of their findings. Although the title of section 7 is “The Localisation of the mammalian Retinal CRB complex”, there is no localization information about other components of the Crb complex. The authors only describe the localization of Crb 1 and 2 and, to a lesser extent, of 3. Do other components in the complex show a similar localization? It may be useful to describe other components of the Crb complex in human if any information is available. Figure 6 does not illustrate the localization of Crb1 and 2 in retinal progenitor cells and RPE. Since expression of Crb1 and 2 differs in human eye, at least in terms of timing, it would be helpful to visualize their localization during development. Inclusion of the protein structure of Crb1, 2 and 3 would help to explain their overlapping function and compensation. It would also be informative to show where the mutations mentioned in the retinal organoid section are located in the protein structures. It is difficult to grasp the phenotypic differences among cell type specific deletion of Crb2 in the Crb1 KO background based on the information presented in the section: a table or figure would be helpful. How do the spatial and temporal aspects of rosette formation differ in cell type specific deletion mutants? The review also never clearly states the differences or distinctive features of LCA-like and RP-like, and, although it also mentions early onset RP, it does not describe the difference between early onset RP and LCA.

Minor points

Line 481, “Binding of PAR6 to the C-terminal PDZ domain of CRB leads to the recruitment of the other PAR complex”

-As CRB does not have a PDZ domain, “PDZ domain” should be changed to “PDZ binding domain”

Line 715 “There may be a common mechanism of active AAV uptake to photoreceptors through there segments, e.g.” 

- “there” should be changed to “their”

Line 649 “CRB2 proteins in the superior versus inferior retina as well as well as the non-uniform localisation of”

-Delete “as well”

Line 604 retinas in which we detected ectopic cells in the ganglion cell layer either sporadically (Crb1KOCrb2ΔRPC), at the peripheral retina (Crb1KOCrb2ΔMG), or within most of the retina (Crb1KOCrb2ΔimPRC) or throughout the retina (Crb1KOCrb2ΔRPC).

Delete underlined (Crb1KOCrb2ΔRPC) or clarify if it is meant to be there.

Author Response

Reviewer 2

In this manuscript, after first presenting broad accounts of visual system structure and function and retinal development and describing a retinal organoid model system that has emerged relatively recently, its main focus, the Crb protein complex and its interacting partners are introduced. It then summarizes Crb1 and 2 protein distribution, primarily in the OLM of Muller glia and photoreceptors of mice, non-human primates and humans.  Subsequent sections describe findings from retinal organoids derived from human patient iPSCs carrying missense mutations of Crb1 and consider the pathogenic mechanisms of Crb1/2 based on studies of mouse genetic models.  To support their contention that Crb2 is a major contributing factor to the variability in onset and severity of Crb1 LCA and RP pathogenesis, the authors then summarize their previous work using various Cre drivers to delete Crb2 in the Crb1 null background. They conclude by addressing the utility of gene therapy, which is a prominent and promising therapeutic approach to retinal dystrophy of genetic origin, in Crb1 patients. Because the authors’ lab has accumulated experimental data and published extensively in this field of research, the review’s focus is primarily on the authors’ own findings. Overall, the review includes a wide array of information about interesting disease-relevant genes, genetic and in vitro models, and therapeutic approaches. The manuscript can be improved by better organization of some paragraphs and clarification of some of the following issues. 

Major points:

In Figure 4, what is the significance that G1 cells after mitosis (right side of cells in the mitosis) do not have apical endfeet? Does it expressly indicate that cells are out of the cell cycle (G0) or that cells transiently lose their apical endfeet during interkinetic migration? The authors should clarify this point.

This was an error. They were meant to be at apical surface. Have corrected.

The first section about retinogenesis primarily focuses on progenitor cell division and fate acquisition, which should be separated from migration of postmitotic retinal cells. Therefore, the tangential migration portion of Figure 4 may be combined with the radial migration shown in Figure 5; this change would improve the organization of these Figures and the section.

Changed as suggested.

Section 6, titled “The Apical CRB and PAR Complexes,” would be improved significantly by reorganizing and clarifying whether the findings apply to other tissue or epithelial cell culture or to the retinal cells. It would be best to first describe the cell biological significance of the interaction between Crbs and binding partners and then describe the relevance to retinal biology and retinal dystrophy. Some animal models, such as Nr2e3rd7 and Nrl-/- mice, do not have sufficient information regarding their genetic cause or phenotype, which makes it difficult to understand the meaning of their findings.

To further clarify this text, additional description was provided to clarify what information was from other tissues or epithelial cell culture or retinal cells if not already done so.

Although the title of section 7 is “The Localisation of the mammalian Retinal CRB complex”, there is no localization information about other components of the Crb complex. The authors only describe the localization of Crb 1 and 2 and, to a lesser extent, of 3. Do other components in the complex show a similar localization? It may be useful to describe other components of the Crb complex in human if any information is available.

Added as suggested.

Figure 6 does not illustrate the localization of Crb1 and 2 in retinal progenitor cells and RPE. Since expression of Crb1 and 2 differs in human eye, at least in terms of timing, it would be helpful to visualize their localization during development.

Added as suggested, to figure 7. RPE expression mentioned in text.

Inclusion of the protein structure of Crb1, 2 and 3 would help to explain their overlapping function and compensation. It would also be informative to show where the mutations mentioned in the retinal organoid section are located in the protein structures.

Added as suggested.

It is difficult to grasp the phenotypic differences among cell type specific deletion of Crb2 in the Crb1 KO background based on the information presented in the section: a table or figure would be helpful.

Added additional information as suggested.

How do the spatial and temporal aspects of rosette formation differ in cell type specific deletion mutants?

This has not been investigated, in sufficient detail to comment.

The review also never clearly states the differences or distinctive features of LCA-like and RP-like, and, although it also mentions early onset RP, it does not describe the difference between early onset RP and LCA.

Added additional information as suggested.

Minor points

Line 481, “Binding of PAR6 to the C-terminal PDZ domain of CRB leads to the recruitment of the other PAR complex”-As CRB does not have a PDZ domain, “PDZ domain” should be changed to “PDZ binding domain”

Changed as suggested.

Line 715 “There may be a common mechanism of active AAV uptake to photoreceptors through there segments, e.g.” - “there” should be changed to “their”

Changed as suggested.

Line 649 “CRB2 proteins in the superior versus inferior retina as well as well as the non-uniform localisation of” -Delete “as well”

Changed as suggested.

Line 604 retinas in which we detected ectopic cells in the ganglion cell layer either sporadically (Crb1KOCrb2ΔRPC), at the peripheral retina (Crb1KOCrb2ΔMG), or within most of the retina (Crb1KOCrb2ΔimPRC) or throughout the retina (Crb1KOCrb2ΔRPC).-Delete underlined (Crb1KOCrb2ΔRPC) or clarify if it is meant to be there.

Changed as suggested.